# The Role of NF-κB in Physiological Bone Development and Inflammatory Bone Diseases: Is NF-κB Inhibition “Killing Two Birds with One Stone”?

**DOI:** 10.3390/cells8121636

**Published:** 2019-12-14

**Authors:** Eijiro Jimi, Nana Takakura, Fumitaka Hiura, Ichiro Nakamura, Shizu Hirata-Tsuchiya

**Affiliations:** 1Oral Health/Brain Health/Total Health Research Center, Faculty of Dental Science, Kyushu University, 3-1-1 Maidashi, Higashi-ku, Fukuoka 812-8582, Japan; 2Laboratory of Molecular and Cellular Biochemistry, Faculty of Dental Science, Kyushu University, 3-1-1 Maidashi, Higashi-ku, Fukuoka 812-8582, Japan; r15takakura@fa.kyu-dent.ac.jp (N.T.); hifumi@dent.kyushu-u.ac.jp (F.H.); 3Faculty of Health and Medical Science, Teikyo Heisei University, 2-51-4 Higashi-Ikebukuro, Toshima, Tokyo 170-8445, Japan; i.nakamura@thu.ac.jp; 4Department of Biological Endodontics, Graduate School of Biomedical and Health Sciences, Hiroshima University, Hiroshima 734-8553, Japan; shtsuchiya@hiroshima-u.ac.jp

**Keywords:** NF-κB, inflammation, osteoclasts, osteoblasts

## Abstract

Nuclear factor-κB (NF-κB) is a transcription factor that regulates the expression of various genes involved in inflammation and the immune response. The activation of NF-κB occurs via two pathways: inflammatory cytokines, such as TNF-α and IL-1β, activate the “classical pathway”, and cytokines involved in lymph node formation, such as CD40L, activate the “alternative pathway”. NF-κB1 (p50) and NF-κB2 (p52) double-knockout mice exhibited severe osteopetrosis due to the total lack of osteoclasts, suggesting that NF-κB activation is required for osteoclast differentiation. These results indicate that NF-κB may be a therapeutic target for inflammatory bone diseases, such as rheumatoid arthritis and periodontal disease. On the other hand, mice that express the dominant negative form of IκB kinase (IKK)-β specifically in osteoblasts exhibited increased bone mass, but there was no change in osteoclast numbers. Therefore, inhibition of NF-κB is thought to promote bone formation. Taken together, the inhibition of NF-κB leads to “killing two birds with one stone”: it suppresses bone resorption and promotes bone formation. This review describes the role of NF-κB in physiological bone metabolism, pathologic bone destruction, and bone regeneration.

## 1. Introduction

As well as supporting and protecting the body, bones support movement in coordination with muscles, host hematopoiesis, and store minerals such as calcium [1,2]. Although bone appears to be a static tissue, it is, in fact, a dynamic tissue that is resorbed and formed constantly and repeatedly. This is called “bone remodeling” [1,2]. Osteoclasts that differentiate from hematopoietic stem cells are responsible for bone resorption, and osteoblasts that differentiate from mesenchymal cells are responsible for bone formation [1,2,3,4,5]. The differentiation and function of these cells are tightly regulated by hormones and local cytokines. In normal bone, the balance between bone resorption and bone formation is maintained, and the bone mass is kept constant; however, in inflammatory bone diseases, such as osteoporosis, rheumatoid arthritis (RA), and periodontitis, bone resorption exceeds bone formation, and then the balance is lost [1,2,3,4,5].

Excessive immune and inflammatory responses enhance bone resorption by osteoclasts and cause bone destruction by impairing osteoblastic bone formation [6]. RA, a typical inflammatory bone disease, is characterized by chronic polysynovitis accompanied with bone destruction by systemic autoimmunity [6]. During the RA activity period, the pannus, which is an abnormal layer of fibrovascular or granulation tissue, infiltrates into the destroyed joint by osteoclasts. At this time, the T-cell immune reaction in the RA synovium causes an excessive biological reaction, and the signal in the synovial cells is continually activated. As a result, synovial cells produce inflammatory cytokines, such as interleukin (IL)-1, IL-6, and tumor necrosis factor (TNF)-α, and matrix-degrading enzymes, such as matrix metalloproteases (MMPs) [7,8].

Nuclear factor-κB (NF-κB) is a collective term for five transcription factors: p50/p105, p52/p100, p65 (relA), c-Rel, and RelB, which together form homo- or heterodimers [9,10]. NF-κB binds to the inhibitory molecules IκBs, IκBα, IκBβ, IκBγ, and IκBε, and they remain as a complex sequestered in the cytoplasm. When cells are stimulated by inflammatory cytokines such as TNF-α and IL-1, IκB is phosphorylated by the IκB kinase complex (IKKα, IKKβ, NEMO: NF-κB essential modulator), ubiquitinated, and then degraded by the ubiquitin-proteasome system [9,10,11]. Then, free NF-κB translocates into the nucleus, recognizes specific DNA sequences, and binds to them to regulate the expression of target genes. This is called “the classical NF-κB pathway”. However, there is also an NF-κB activation mechanism that is independent of IκB degradation. In the unstimulated state, p100 remains in the cytoplasm by associating with RelB. When activated, the C-terminal end of p100, which has the same function as IκBs, is degraded, and then a heterodimer of RelB/p52 is formed and translocates into the nucleus. This activation pathway is referred to as “the alternative NF-κB pathway” (Figure 1) [9,10,11]. Since these two pathways play different roles, the p50/p65, p50/c-Rel, and p52/RelB heterodimers are expected to bind to their specific DNA sequences. However, the sequence to which the p52/RelB heterodimer specifically binds and its target genes have not been identified [11,12] (Figure 1).

NF-κB is a transcription factor regulated by genes that control immune and inflammatory responses, but NF-κB1 (p50) and NF-κB2 (p52) double-knockout (dKO) mice exhibited severe osteopetrosis and lacked osteoclasts, suggesting that NF-κB also directly controls osteoclast differentiation [13,14]. Recent findings also showed that NF-κB controls osteoblast differentiation directly or indirectly [15,16]. In this review, we will mainly explain the physiological and pathological roles of NF-κB in bone development and disease, focusing on osteoclasts and osteoblasts.

## 2. The Role of NF-κB on Bone Metabolism

### 2.1. The Functions of NF-κB Signaling in Physiological Osteoclastic Bone Resorption

Osteoclasts differentiate from hematopoietic stem cells into osteoclasts via macrophage and monocyte pathways [3,4,5,17]. During differentiation, osteoclast progenitor cells proliferate and differentiate into mono- and binucleated osteoclasts that fuse to become multinucleated osteoclasts. Multinucleated osteoclasts recognize the bone matrix, form a sealing zone to separate the resorption surface from the outside, form a ruffled border, and secrete acid and proteolytic enzymes into the resorption lacunae [3,4,5,17].

Osteoclast differentiation is controlled by two cytokines: macrophage colony-stimulating factor (M-CSF) and receptor activator of NF-κB ligand (RANKL). M-CSF is essential for differentiation into osteoclast progenitors, and it induces the expression of the RANKL receptor RANK. Transcription factors PU.1 and MITF induce the expression of M-CSF receptor (*c-fms*), and individuals lacking these transcription factors have impaired osteoclast differentiation and present with marble bone disease. In addition, *op*/*op* mice and *c-fms*-deficient mice that cannot produce functional M-CSF exhibit marble bone disease and lack osteoclasts [3,4,5,17].

RANKL is produced by various cells, such as osteoblasts, osteocytes, T cells and B cells [3,4,5,17]. Mice that lack RANKL and its receptor RANK have severe osteopetrosis caused by a total lack of osteoclasts [18,19]. On the other hand, the number of osteoclasts increases in mice that lack the RANKL decoy receptor, osteoprotegerin (OPG), resulting in osteoporosis [20,21]. In human hereditary bone disease, mutations in RANKL, RANK, and OPG have been found, and these three molecules have been shown to be important for osteoclast formation, which maintains bone mass [22].

RANK belongs to the TNF receptor family, and various adapter molecules can interact with the intracellular domain of RANK [23]. Among TNF receptor-activating factor (TRAF) members, TRAF6-deficient mice exhibit an osteopetrosis phenotype that is similar to RANKL- or RANK-deficient mice [24]. Of the downstream molecules of TRAF6, c-Fos and c-Jun regulate the transcription factor AP-1. c-Fos-deficient mice also exhibited osteopetrosis [25]. Another downstream molecule, the transcription factor NF-κB, is composed of five family members. In mice with both NF-κB1 and NF-κB2 knocked out, there is also osteopetrosis due to the total lack of osteoclasts, but deletion of either NF-κB1 or NF-κB2 alone causes no detectable bone phenotype [13,14]. The molecular mechanism by which osteoclasts cannot form in NF-κB1 and NF-κB2 dKO mice is still unknown, but it is certain that NF-κB signaling is important for osteoclast formation.

Among the molecules involved in the signal transduction of NF-κB, p65 (RelA), IKKβ and NEMO could not be analyzed regarding a bone phenotype because these molecules are embryonic lethal [26,27,28,29,30,31]. Thus, to make IKKβ specifically deficient (IKKβcKO) in myeloid cells, IKKβ^flox/flox^ mice were crossed with Mx1 or CD11b-Cre transgenic mice to generate conditional knockout mice in which IKKβ is specifically deficient (IKKβcKO) in myeloid cells [32,33]. IKKβcKO mice showed an increase in the trabecular bone volume due to a decrease in the number of osteoclasts. Furthermore, the number of osteoclast precursor cells (F4/80 positive cells) was also significantly reduced. When IKKβcKO mice were crossed with tumor necrosis factor receptor 1(TNFR1)^–/–^ mice to generate IKKβcKO/TNFR1KO dKO mice, osteoclast precursor cells were resistant to apoptosis; further, IκBα was not degraded by RANKL stimulation, and osteoclast differentiation was still suppressed. By contrast, in IKKα knock-in (IKKα^A/A^) mice in which the serine residue necessary for IKKα kinase activity was substituted with alanine, osteoclast formation by RANKL stimulation was suppressed in vitro but not in vivo. The trabecular bone volume in IKKα^A/A^ mice was comparable to that of wild-type (WT) mice [30]. Furthermore, IKKβ-deficient osteoclasts resulted in RANKL-induced apoptosis by the activation of c-Jun N-terminal kinase (JNK), and the addition of JNK inhibitor restored RANKL-induced apoptosis derived from IKKβcKO mice in vitro [33]. Thus, IKKβ, but not IKKα, is important as a RANK downstream signal in osteoclast differentiation. Consistent with these results, treatment with specific inhibitors of IKKβ activity suppressed RANKL-induced osteoclastogenesis in vitro and in vivo [34,35,36,37].

Since p65-deficient (p65^–/–^) mice are also embryonic lethal, p65^–/–^ fetal liver cells were studied; the cells were transplanted into irradiated mice to reconstitute bone marrow cells. Fewer osteoclasts were observed in p65^–/–^ chimera mice. When p65^–/–^ chimera mice were crossed with TNFR1^–/–^ mice, p65^–/–^ precursors were found to be sensitive to RANKL-induced apoptosis even on the TNFR1^–/–^ background. ZVAD, a caspase inhibitor, restored RANKL-induced osteoclastogenesis in p65^–/–^ precursors in vitro, suggesting that p65 induces proapoptotic gene expression in osteoclastogenesis [38].

Several lines of evidence have shown that the alternative NF-κB pathway also involves RANKL-induced osteoclastogenesis. When RANKL is administered to NIK-deficient (NIK^–/–^) mice, osteoclast formation is more inhibited than it is when RANKL is administered to wild-type mice. Osteoclast progenitor cells derived from NIK^–/–^ mice did not induce processing of p100 to p52 by RANKL stimulation due to IκB-like function of the C-terminus of p100 [39,40]. In addition, mice lacking IKKα, which is a molecule that is downstream of NIK, contain osteoclasts but are small in size and have reduced bone resorbing activity. As with NIK^–/–^ mice, processing of p100 to p52 by RANKL stimulation does not occur in IKKα-deficient mice [32,41]. The role of RelB in osteoclastic bone resorption is still unclear. Although the number of osteoclasts was normal, the bone mass was slightly increased. However, overexpression of RelB restored RANKL-induced osteoclastogenesis in NIK^–/–^ mice [42]. Recently, NIK-deficient and RelB-deficient female mice, but not male mice, revealed a 2-fold increase in trabecular bone mass, suggesting that the alternative NF-κB pathway involves gender difference in bone metabolism [43]. Alymphoplasia (*aly/aly*) mice do not undergo p100 to p52 processing because NIK is inactive. *Aly/aly* mice showed mild osteopetrosis and had a greatly reduced osteoclast count [44,45]. RANKL-induced osteoclast formation from the bone marrow cells of *aly/aly* mice was also suppressed. RANKL still induced IκBα degradation and activated classical NF-κB, but p100 to p52 processing was abolished by the *aly/aly* mutations. Overexpression of NFATc1 and constitutive activation of IKKα or p52 restored RANKL-induced osteoclastogenesis in *aly/aly* cells. The overexpression of RelB in *aly/aly* cells restored RANKL-induced osteoclastogenesis by inducing cancer Osaka thyroid (Cot) expression, which induces the processing of p52 from p100 in place of NIK [46]. Taken together, the balance between p52 and p100 determines RANKL-induced osteoclastogenesis.

### 2.2. NF-κB Inhibition Suppresses Inflammatory Bone Diseases

#### 2.2.1. Rheumatoid Arthritis (RA)

Rheumatoid arthritis (RA) is a chronic inflammatory disease with progressive joint destruction over time [6,7,8]. Biologics such as anti-TNF-α antibodies have been shown to be effective in cases where existing drugs have not been effective [47]. The characteristic feature of RA is the proliferation and infiltration of synovial cells and angiogenesis of the joint area [6,7,8]. In the joint area, the overproduction of inflammatory cytokines such as IL-1, TNF-α, IL-6, and IL-17, adhesion molecules, and MMPs and the induction of osteoclasts are involved in bone and cartilage destruction in RA [6,7,8]. Recently, biological products, such as anti-TNF-α neutralized antibody (etanercept, infliximab, and adalimumab, etc.,) and anti-IL-6 neutralized antibody (tocilizumab), which are drugs created by biotechnology, have been used for rheumatoid arthritis. Compared to conventional antirheumatic drugs, the cost is high, but it is known to be particularly effective in suppressing joint destruction. Treatment guidelines exist to prevent the destruction of joints by introducing biologics as soon as possible when treatments centered on rheumatox are not enough to control the disease. These guidelines are widely accepted internationally [47]. Anti-TNF-α neutralized antibodies directly inhibit the binding of TNF-α to its receptor and suppress excessive inflammation that induces RANKL expression in synovial cells. IL-6 is required for the differentiation of Th17 cells that promote osteoclast differentiation, and these neutralizing antibodies are thought to not only sink local inflammation but also suppress RANKL induction and osteoclast differentiation. However, these biologics cause serious side effects, such as triggering an autoimmune anti-antibody response or weakening the body’s immune defenses. Therefore, alternative small-molecule-based therapies for inhibition of these cytokines’ effects is a hot topic both in academia and industry [47,48].

Since NF-κB is a transcription factor that regulates the expression of inflammatory cytokines, including TNF-α and IL-6, and serves as mediator for RANK signaling, selective inhibition of the classical NF-κB pathway appears to be a target for RA bone destruction [9,10,11]. Thus, to suppress the classical NF-κB pathway, experiments have been conducted [34,35,49,50,51,52,53,54] on the treatment of arthritis models with NF-κB inhibitors, such as decoy oligonucleotides, NEMO-binding domain (NBD) peptide, TAT-IκBα-super repressor, the dominant negative form of IKKβ, or IKKβ inhibitors such as N-(6-chloro-7-methoxy-9H-beta-carbolin-8-yl)-2-methylnicotinamide (ML120B), 4(2′-aminoethyl)amino-1,8-dimethylimidazo(1,2-a)quinoxaline (BMS-345541), 2-methoxy-N-((6-(1-methyl-4-(methylamino)-1,6-dihydroimidazo[4,5-d]pyrrolo[2–b]pyridin-7-yl)pyridin-2-yl)methyl)acetamide (BMS-066), or (7-[2-(cyclopropyl-methoxy)-6-hydroxyphenyl]-5-[(3*S*)-3-piperidinyl]-1,4-dihydro-2*H*-pyrido[2–*d*][1,3]oxazin-2-one hydrochloride (CHPD). These inhibitors can suppress bone destruction by suppressing local inflammation and osteoclast formation (Figure 2).

Iguratimod (IGU), a methanesulfonanilide, is a novel disease-modifying antirheumatic drug (DMARD) that inhibits the production of immunoglobulins without affecting B cell proliferation, various inflammatory cytokines (IL-1, -6 and -8 and TNF-α), and osteoclastogenesis by inhibiting NF-κB [55]. IGU is orally bioavailable and easily absorbed from the gastrointestinal tract, and food does not affect its pharmacokinetics. Several clinical studies have shown that IGU has immediate and long-lasting effects on RA treatment [55,56]. Thus, IGU has been acceptable as an alternative where other DMARDs are less effective or conventional RA treatment does not work well. However, some side effects, including nausea, dizziness, headaches, and itching, have been reported [55]. A recent study identified that patients carrying the *ABCG2 rs2231142* allele were highly responsive to IGU, while those carrying *NAT rs1495742G* had the lowest response. Furthermore, patients carrying *CYP2C19*2 rs4244285* had a higher risk of IGU toxicity [57]. This report may useful to predict the patient’s response to IGU and to avoid the potential toxicity.

It has been reported that not only these inhibitors but also components contained in plant extracts, such as turmeric supplements, *Trachelospermi caulis*, *Moutan cortex radicis*, or *Saposhnikovia divaricata*, can suppress the activation of the classical NF-κB pathway, which mediates excessive immune responses and inflammation followed by cartilage destruction in arthritis [58,59,60,61,62,63,64,65,66]. The dietary ω-3 polyunsaturated fatty acids (PUFAs), eicosapentaenoic acid (EPA) and docosahexaenoic acid (DHA), originating from fish oils, also reduce pain and inflammation in RA by suppressing IL-1 or TNF-α production via NF-κB activation. A recent clinical trial indicated that when fish oil was used as an adjunctive therapy in drug treatment for recent onset RA, rates of remission increased and drug use decreased [67]. Moreover, a daily diet of extra-virgin olive oil significantly reduced joint edema and cartilage destruction, preventing arthritis development in a mouse CIA model by suppressing inflammatory cytokines and MMP3 production induced by the Janus kinase signal transducer and activator of transcription (JAK/STAT), mitogen-activated protein kinases (MAPKs), and the NF-κB pathway [68]. These ingredients are safe because they can be taken into the body as foods and supplements, but they are less effective as therapeutic agents for rheumatism, and may be synergistic when used as an aid to therapeutic agents.

Recent findings showed the involvement of the alternative NF-κB pathway on the development of RA [69,70]. NIK is highly expressed in synovial endothelial cells of RA patients [69]. NIK promotes pathogenic angiogenesis and synovial inflammation via CXCL12 production from endothelial cells [70]. Furthermore, NIK^–/–^ mice have been found to be resistant to antigen-induced arthritis resulting from T cell responses [40,71]. So far, there is no suggestion that the specific inhibitor of NIK suppresses bone destruction on an RA model. However, for the B cell activating factor belonging to the tumor necrosis factor family (BAFF), which activates the alternative NF-κB pathway [10], antagonists improved the arthritis score of collagen-induced arthritis [72]. Taken together, these findings show that the alternative pathway is involved in the development of RA.

#### 2.2.2. Ankylosing Spondylitis (AS)

Ankylosing spondylitis (AS) is a chronic arthritis accompanied by inflammation of bone at the cartilage–bone interface [73,74]. AS develops with time via chronic inflammation mainly in the spine, and extra bone is formed in the spine, followed by the fusion of vertebrae. The disease’s most prominent onset starts from the ages of 20 to 30 and is most prominent in males; men and women are affected at a ratio of approximately 3:1. Although the etiology of AS is unknown, HLA-B27 belongs to the class-1 surface antigens present on the interface of “MHC” antigenic peptides of T-cells, and is mainly involved in the pathogenesis of AS. The functions of HLA-B27 regulate its ability to misfold, to induce an endoplasmic reticulum stress response, and to promote autophagy/unfolded protein responses (UPR). The expression of UPR genes induces inflammatory cytokine production, such as TNF-α and IL-17 from Th17 cells [73,74]. Since AS is an HLA-B27-linked inflammatory disease, AS has been treated with anti-inflammatory or immunosuppressive drugs [73]. Recent data suggested a role for TNF-α in the pathophysiology of AS and showed that TNF-α mRNA is upregulated in the sacroiliac joints of AS patients. Therefore, if a patient continues to suffer high AS disease symptoms and the conventional treatments are not effective, then anti-TNF-α or anti-TNF receptor antibodies (such as adalimumab, etanercept, certolizumab pegol, infliximab, and golimumab) can be an option [75]. Anti-TNF antibodies not only effectively treat AS but they also suppress inflammation and improve spine mobility with sustained effects. In general, suppressing TNF not only suppresses bone resorption but also enhances bone formation [76]. AS is a disease in which excessive bone formation occurs, but an improvement in symptoms is considered to be an anti-inflammatory effect that is stronger than the promotion of bone formation. However, it is necessary to select alternative DMARDs, including selective NF-κB inhibitor, for patients who do not respond well to TNF-α inhibition, or when considering the costs and side effects of anti-TNF-α treatments.

#### 2.2.3. Periodontal Disease

Periodontal disease is a chronic inflammation caused by a bacterial infection [77,78]. Bacterial plaques induce host inflammation, and the ongoing inflammatory response induces periodontal tissue destruction. Periodontal disease is characterized by the formation of periodontal pockets, the resorption of alveolar bone, and the movement of the tip barrier of the epithelium, which destroys periodontal tissue. It is also well known that multiple risk factors accelerate periodontitis. Periodontal disease is mainly treated by the mechanical removal of causative substances such as bacteria and plaque; pharmacotherapy is not very effective. This difficulty in treatment may be because of the anatomical complexity of the periodontal tissue and its constant contact with the external environment [77,78]. Since NF-κB is involved in the onset and progression of various inflammatory diseases, pharmacotherapies targeting NF-κB have been attempted. Similarly, in periodontal disease, the administration of IMD-0354, a novel NF-κB inhibitor that suppresses IKKβ activity, has been used in ligation-induced periodontitis models, and it significantly suppresses RANKL, IL-1β, and TNF-α expression in gingival tissues. Furthermore, the number of osteoclasts also decreased following treatment, and bone resorption was suppressed [79]. The application of NF-κB inhibitors may represent new pharmacotherapy options for periodontal patients.

### 2.3. The Activation of NF-κB Suppresses Bone Formation

Bone is composed of hydroxyapatite crystals and various extracellular matrix proteins, including type I collagen, osteocalcin, osteopontin, bone sialoprotein, and proteoglycan. Most of these bone matrix proteins are secreted and deposited by mature osteoblasts that are aligned on the bone surface. The formation of hydroxyapatite crystals in osteoid is also regulated by osteoblasts. The expression of numerous bone-related extracellular matrix proteins and the activity of alkaline phosphatase (ALP) are key features of osteoblasts [1,2].

Osteoblasts differentiate from mesenchymal stem cells, and their differentiation stage is cooperatively and dynamically controlled by specific signal transduction pathways, directly or indirectly. Osteoblasts differentiate from mesenchymal stem cells through various intracellular signaling mechanisms by various cytokines and hormones, such as bone morphogenetic proteins (BMPs), transforming growth factor (TGF)-β, Wnt, hedgehog, fibroblast growth factor, and estrogen. This intracellular signal transduction is activated by phosphorylation, ubiquitination, protein–protein interactions and structural changes following the binding of ligands to receptors. Since mice with either Runx2 or Osterix transcription factors knocked out exhibited impaired bone formation, these two transcription factors have been reported to be important for osteoblast differentiation [1,2,80].

It is known that bone formation is suppressed in an inflammatory state, and, in particular, TNF-α is known to suppress osteoblast differentiation in various culture systems [76,81,82,83,84]. TNF-α activates various signals in the cell, but a specific IKK inhibitor, BAY11-7082, restores the suppression of osteoblast differentiation induced by TNF-α [85]. Recently, it has been reported that the inhibition of NF-κB by the dominant negative form of IKKβ enhances bone formation [13]. The administration of an inhibitor of IKK, S1627, promoted bone formation in ovariectomized (OVX) mice [14]. Mice expressing the dominant negative form of IKKβ in mature osteoblasts showed increased bone mass, bone mineral density, and osteoblast activity without exhibiting any changes to osteoclast activity. Furthermore, expressing the dominant negative form of IKKβ maintained the bone mass of OVX mice by increasing the expression of Fos-related antigen-1 (Fra1), which is an essential transcription factor involved in bone matrix formation [13]. There are also reports supporting these findings that show that estrogen receptors inhibit the activation of the classical NF-κB pathway by interacting with NF-κB [86]. As another possibility, TNF-α, IL-1β, IL-6, and IL-17 produced by T cells and other cells during osteoporosis have been reported to activate the classical NF-κB pathway [87].

Bone morphogenetic proteins (BMPs) belong to the TGF-β superfamily and were originally identified by their ability to induce ectopic bone formation when implanted into muscle tissue [88,89]. Since BMP signaling and the classical NF-κB pathway have opposing biological activities, crosstalk between the two is possible. A cell-permeable inhibitor of the classical NF-κB pathway restored the inhibitory effects of TNF-α on BMP2-induced Runx2 expression and osteoblast differentiation [90]. Zinc inhibits the classical NF-κB pathway by TNF-α and promotes BMP2-induced osteoblast differentiation [91]. Pyrrolidine dithiocarbamate (PDTC), which inhibits the classical NF-κB pathway, partially blocked the TNF-α-induced suppression of osteoblast differentiation. These results indicate that inhibition of the classical NF-κB pathway by BMP [92] reverses osteoblast differentiation in a mechanism dependent on TNF-α. Thus, the implantation of collagen sponges containing BAY11-7082, a selective inhibitor of the classical NF-κB pathway, with BMP2 under the fascia resulted in the formation of larger amounts of ectopic bone than what was seen following treatment with only BMP2 [93]. These results suggest that selective inhibitors of the classical NF-κB pathway have the effect of promoting bone formation by BMP. However, the side effects of its administration must be considered, since inhibition of the classical NF-κB pathway activity might induce cell death [26,27,28,29,30,31]. Therefore, to enhance the effect of BMP without impairing the function of the classical NF-κB pathway, the suppression mechanism of the BMP/Smad signal of the classical NF-κB pathway was examined. There are various stages of BMP/Smad signaling, but the classical NF-κB pathway does not affect the phosphorylation of Smad1/5 or the formation of the Smad1–Smad4 complex; however, the classical NF-κB pathway does interfere with the DNA binding complex. Furthermore, we found that the p65 subunit of the classical NF-κB pathway associates with Smad4 but not Smad1 [64]. Therefore, when the association sites of p65 and Smad4 were examined, the transactivation domain 2 (TA2) of p65 and the mad homology (MH) 1 region of Smad1 were directly associated. We further narrowed the association site to the amino acid level and found that the 16 amino acid sites on the N-terminal side of the TA domain of p65 were critical for binding to Smad4; we named the site the Smad binding domain (SBD) [94]. We synthesized the SBD peptide to compete with the interaction of p65 with Smad4. The SBD peptide promoted ALP activity and calcification induced by BMP2 in vitro. Furthermore, administration of SBD peptide together with BMP2 induced ectopic thick cortical bone formation in vivo. The SBD peptide did not affect the activation of the classical NF-κB pathway by TNF-α stimulation [94]. Based on these results, it is possible that peptides targeting the association site of NF-κB, p65 and Smad4 may be useful for promoting bone formation by BMP with few side effects (Figure 3).

Recently, a heterozygous de novo missense mutation (c.1534_1535delinsAG, p.Asp512Ser) in exon 11 of RELA encoding Rela/p65 was found in a neonate who had died suddenly and unexpectedly with high bone mass (HBM) that was judged radiographically and by skeletal histopathology [95]. Numerous morphologically normal osteoclasts in the neonate were observed in bone histology, suggesting that the missense change was associated with neonatal osteosclerosis from increased osteoblastic bone formation in utero rather than failed osteoclastic bone resorption. Moreover, LPS stimulation failed to activate the classical NF-κB pathway in fibroblasts derived from the neonate. This is the first report that demonstrates the importance of the Rela/p65 subunit within the classical NF-κB pathway for human skeletal homeostasis and represents a new genetic cause of HBM [95].

Bone histomorphometric data from *aly/aly* mice show an increase in trabecular bone volume caused by both the suppression of bone resorption and increased bone formation, suggesting that the alternative NF-κB pathway also regulates osteoblastic bone formation [96]. ALP activity and the expression of osteoblastic markers (including osteocalcin, Id1, Osterix, and Runx2) induced by either β-glycerophosphate and ascorbic acid or BMPs were increased in primary osteoblasts (POB) derived from *aly/aly* mice compared with WT mice. The ectopic bone formation in vivo induced by BMP2 was enhanced in *aly/aly* mice compared with WT mice, due to enhancement of BMP2 signaling [96]. Thus, the alternative NF-κB pathway via the processing of p52 from p100 negatively regulates osteoblastic differentiation and bone formation by modifying BMP activity. Mice that have RelB, a main subunit of the alternative NF-κB pathway, knocked out develop age-related increased trabecular bone mass associated with increased bone formation [97]. RelB^–/–^ bone marrow stromal cells enhanced osteoblastic differentiation by increasing Runx2 expression. RelB directly bound to the Runx2 promoter to inhibit its activation. Moreover, RelB^–/–^ bone-derived mesenchymal progenitor cells (MPCs) formed bone more rapidly than WT cells after they were injected into a murine bone defect model [97].

Notch is a family of evolutionarily conserved receptors that regulate cell fate, and its signaling plays various important roles in bone metabolism [98]. Notch signaling and the alternative NF-κB pathway were identified as signaling pathways responsible for the inhibitory effects of TNF-α on osteoblastic differentiation. This was done by RNA sequencing and pathway analysis of mesenchymal stem cells using WT and TNF-α transgenic (Tg) mice, a model of RA [99]. Notch inhibitors restored bone loss and osteoblast inhibition in TNF-α Tg mice. The transplantation of fibroblasts from TNF-α Tg mice treated with Notch inhibitors formed more new bone in recipient mice with bone defects. The activation of the alternative NF-κB pathway in a murine pluripotent stem cell line induced RPBjκ and HES1 in a Notch intracellular domain dependent manner (NICD-dependent). TNF-α enhanced the binding of p52/RelB heterodimer to NICD, which induced binding at the RBPjκ site within the Hes1 promoter. Elevated levels of HES1, p52, and RelB were observed in mesenchymal stem cells from RA patients [99]. These results indicate that the inhibition of the alternative NF-κB pathway could reduce age-related bone loss and enhance bone repair as well as inflammation-mediated bone loss.

## 3. Conclusions

The balance between bone resorption and bone formation is important in maintaining bone mass. Bone resorption is enhanced in a state of inflammation, and bone mass is reduced when bone formation is inhibited. Although the molecular mechanisms of osteoclast differentiation, osteoclast activation, osteoblast differentiation, and bone formation have been analyzed in detail, NF-κB is commonly associated with key terms such as “inflammation”, “bone resorption”, and “suppression of bone formation” [9,10,11]. The inhibitors of NF-κB have been reported to promote bone formation in addition to anti-inflammatory action and osteoclast formation inhibition. Iguratimod (IGU) is a low molecular weight compound that inhibits the classical NF-κB pathway. It is currently used as one of the therapeutic agents for RA and has been reported to be effective [56]. Thus, targeting NF-κB is effective at maintaining bone mass during inflammation [13,14,15,34,35,36,37,49,50,51,52,53,54,55,56,58,59,60,61,62,63,64,65,66,67,68,69,85,91,92,93,94], and there is a possibility for “killing two birds with one stone”. However, embryonic lethality has been reported in mice where molecules involved in the NF-κB signaling have been knocked out [26,27,28,29,30,31], and it is necessary to consider the possibility of causing serious side effects simply by inhibiting NF-κB. In fact, there are reports of RA patients who are less responsive to IGU and patients who experience adverse side effects. To find practical applications of other NF-κB inhibitors, translational research not only for animal experiments, but also for human applications, will become necessary. It is important to continue to work on basic and clinical research regarding the molecular mechanism of inflammatory bone disease to provide more options to patients.

## Figures and Tables

**Figure 1 cells-08-01636-f001:**
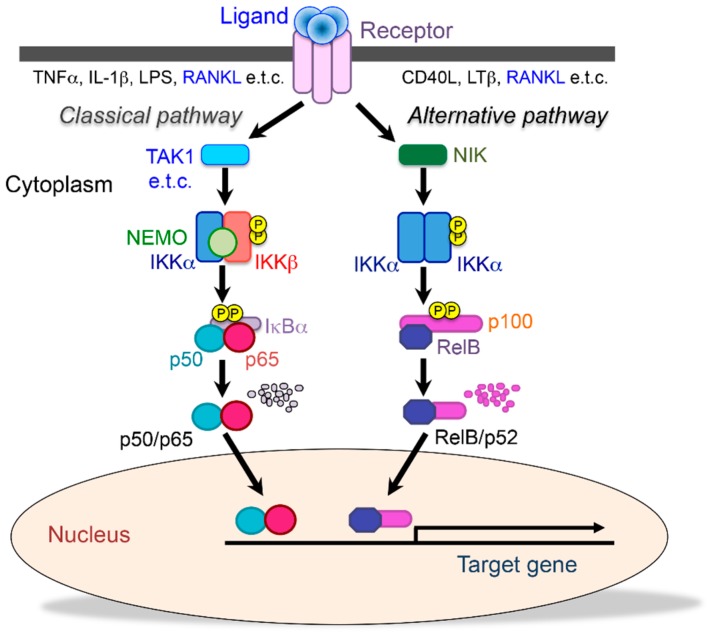
Two different NF-κB signaling pathways. The classical (canonical) pathway (**left**) is activated by a large number of agonists, such as TNF-α, IL-1, lipopolysaccharide, and T cell receptors. The activation of this pathway depends on the IκB kinase (IKK) complex (IKKEMO), which phosphorylates IκBα (Ser32, 36) to induce rapid degradation. This pathway is essential for immune responses, inflammation, tumorigenesis, and cell survival. The alternative (noncanonical) pathway (**right**) is activated by a limited number of agonists, which are involved in secondary lymphoid organogenesis, mature B cell function, and adaptive immunity. This pathway requires NIK and IKKα, which induce the slow processing of p100 to generate p52, resulting in the dimerization and activation of the p52/RelB heterodimer. Receptor activator of NF-κB ligand (RANKL) activates both classical and alternative pathways.

**Figure 2 cells-08-01636-f002:**
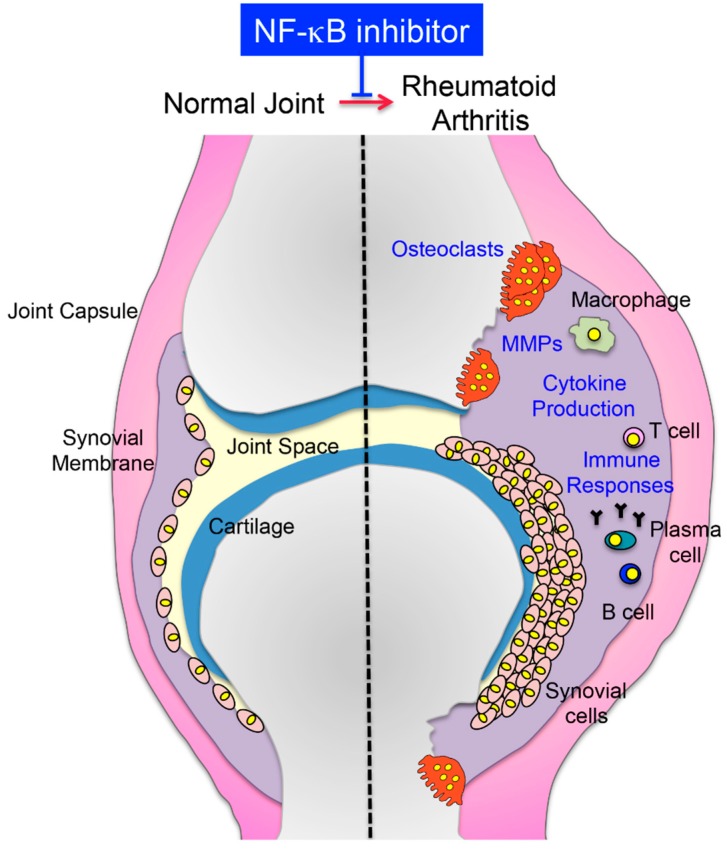
Schematic representation of a healthy joint and rheumatoid arthritis. Rheumatoid arthritis (RA) is a characterized by extensive synovitis, cartilage erosion, and bone destruction by excessive immune and inflammatory responses. Synovial cells and immune cells produce inflammatory cytokines, such as IL-1, IL-6, and TNFα, and matrix metalloproteases (MMPs). NF-κB inhibitors, such as decoy oligonucleotides, NBD peptide, TAT-IκBα-super repressor, or IKKβ inhibitor, suppress bone destruction by suppressing local inflammation.

**Figure 3 cells-08-01636-f003:**
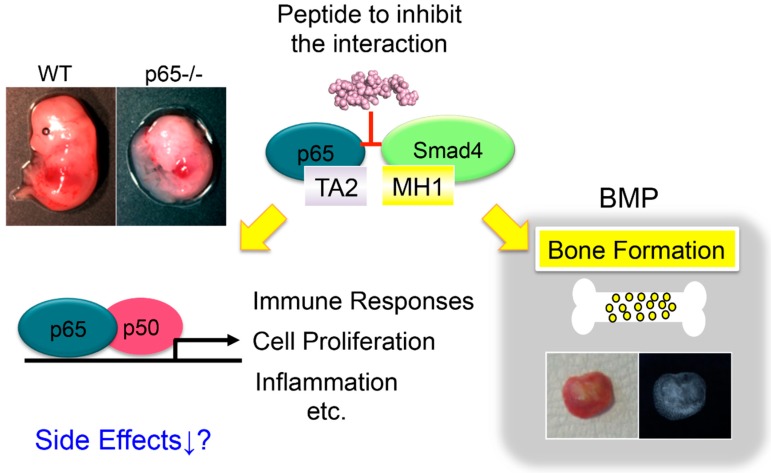
A peptide that blocks the interaction of NF-κB p65 subunit with Smad4 enhances BMP2-induced bone formation. Inhibitors of the classical NF-κB pathway have been reported to promote bone formation, but mice deficient in p65, the main subunit of NF-κB, are embryonic lethal and must be considered for possible side effects. Therefore, we investigated the molecular mechanism by which NF-κB suppresses BMP signaling and revealed that p65 and the BMP-signaling molecule Smad4 are associated. We suggested the possibility of enhancing the effect of BMP without impairing the function of NF-κB by using a peptide that specifically inhibits the association site.

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
