# Peer review of "The Role of NF-κB in Physiological Bone Development and Inflammatory Bone Diseases: Is NF-κB Inhibition “Killing Two Birds with One Stone”?"

_cells, 2019, doi:10.3390/cells8121636_

Round 1

Reviewer 1 Report

This review of NF-kB in osteoclasts and osteoblasts is fairly comprehensive and clearly written, but there are a few areas in which additional publications should be cited for a more complete and balanced view of the field. Additionally, while the 2 distinct pathways are described up front, in several places (detailed below) only data on the classical pathway is included and referred to simply as “NF-kB.”  For clarity, it would be best to continue to use the more specific pathway designations.

Specific comments:

There are random squares on all of the figures that don’t belong Line 44 – word “associated” is used twice and should be revised Line 45 – although the pannus represents an expansion of tissue, its description as “tumor-like” is not very accurate. The destruction of the bone that allows the apparent “invasion” is accomplished by osteoclasts, not the synovium or inflammatory cells there.   Line 60 – the description of the C-terminal end of p100 having the “same function as degraded IkB” is confusing. It would be better to describe the process directly, such as stating its degradation is analogous to the IkBs. Line 109 – the statement about the cells producing RANKL should include references Line 116 – RANK belongs to the TNF receptor family OR RANK is a member of the TNF receptor family In the paragraph beginning on Line 126, another paper describing IKKbeta conditional knockout mice to consider is Otero JBC 283:24546 Lin 155 – The references listed (36 and 37) do not describe the IKKa deficient mice. Need to include Chaisson JBC 279:54841 An additional paper describing phenotypes in NIK and RelB deficient mice that should be included in the paragraph beginning on line 148 Zarei JBMR Plus 3:14 Line 181 – ref 48 is to a paper on osteoarthritis, not RA Line 182 – Fig 2 should be referenced earlier in this paragraph. Natural herbs are not mentioned in the figure. Line 229 – there should not be an “s” on the word osteoid The sentences beginning on lines 233 and 235 are quite redundant There is now data on the alternative NF-kB pathway in osteoblasts that is not at all addressed in this review. Papers that should be considered include Yao JBMR 29:866; Zhang JCI 124:297; Davis JBMR 34:2087 In Figure 3, it is not clear what the words “time-saving” mean/refer to. Greek symbols seem to be missing from the references.

Author Response

To Reviewer #1,

We would like to thank you for your suggestions, which greatly enhanced our revised manuscript.

First of all, in accordance with reviewer 3’s suggestion, we added new description regarding the advantage of NF-κB inhibitors compared with either TNFablockers or IL-6 neutralized antibody for the treatment with RA, and the involvement of NF-κB on the development of ankylosing spondylitis in the revised manuscript. Since we are dentists and working on the basic research, we are not experts in these matters. Thus, from the standpoint of an orthopedic surgeon, Dr. Nakamura reviewed our description. The added part is written in blue. Since, unless Dr. Nakamura’s proper advice, we could not finish the revised manuscript, we added his name as a new co-author in the revised manuscript.

Specific comments:

There are random squares on all of the figures that don’t belong Line 44 – word “associated” is used twice and should be revised.

I could not see any squares on all of the figures in my computer and printed both word and pdf versions. I am not sure what are these, but I converted ppt file to tif file and then attached new all of the figures in the revised manuscript.

In accordance with your suggestion, we rewrote “RA, a typical inflammatory bone disease, is characterized by chronic polysynovitis accompanied with bone destruction by systemic autoimmunity” on page 2, lines 45-47, in the revised manuscript.

Line 45 – although the pannus represents an expansion of tissue, its description as “tumor-like” is not very accurate. The destruction of the bone that allows the apparent “invasion” is accomplished by osteoclasts, not the synovium or inflammatory cells there.  

In accordance with your suggestion, we rewrote “ During the RA activity period, the pannus, which is an abnormal layer of fibrovascular or granulation tissue, infiltrates into the destroyed joint by osteoclasts.” on page 2, lines 47-48, in the revised manuscript.

Line 60 – the description of the C-terminal end of p100 having the “same function as degraded IκB” is confusing. It would be better to describe the process directly, such as stating its degradation is analogous to the IκBs.

In accordance with your suggestion, we rewrote “When activated, the C-terminal end of p100, which has the same function like IκBs, is degraded, and then, a heterodimer of RelB/p52 is formed and translocates into the nucleus.” On page 2, lines 62-65, in the revised manuscript.

Line 109 – the statement about the cells producing RANKL should include references.

In accordance with your suggestion, we added 4 references, #3-5, 17, on page 3, line 115, in the revised manuscript.

Line 116 – RANK belongs to the TNF receptor family OR RANK is a member of the TNF receptor family

In accordance with your suggestion, we rewrote “RANK belongs to the TNF receptor family,” on page 3,line 122, in the revised manuscript.

In the paragraph beginning on Line 126, another paper describing IKKbconditional knockout mice to consider is Otero JBC 283:24546

In accordance with your suggestion, we added a sentence “Furthermore, IKKb-deficient osteoclasts resulted in RANKL-induced apoptosis by activation of c-jun N-terminal kinase(JNK), and addition of JNK inhibitor restored RANKL-induced apoptosis derived from IKKbcKO mice in vitro” on page 4, lines 145-147, and a reference #33, in the revised manuscript.

Lin 155 – The references listed (36 and 37) do not describe the IKKadeficient mice. Need to include Chaisson JBC 279:54841

In accordance with your suggestion, we added a reference #41 on page 4, line 165, in the revised manuscript.

An additional paper describing phenotypes in NIK and RelB deficient mice that should be included in the paragraph beginning on line 148 Zarei JBMR Plus 3:14

In accordance with your suggestion, we added a sentence “Recently, NIK-deficient and RelB-deficient female mice, but not male mice revealed 2-fold increase in trabecular bone mass, suggesting that the alternative NF-kB pathway involves gender difference in bone metabolism” on page 5, lines 168-170, and a reference #43, in the revised manuscript.

Line 181 – ref 48 is to a paper on osteoarthritis, not RA

In accordance with your suggestion, we deleted the reference. We modified a sentence “It has been reported that not only these inhibitors but also components contained in plant extracts, such as turmeric supplements, Trachelospermi caulis, Moutan cortex radicis,orSaposhnikovia divaricata, etc., can suppress the activation of the classical NF-kB pathway, which mediates excess of immune responses and inflammation followed by cartilage destruction in arthritis” on page 7, lines 254-257, and 8 references #58-65 in the revised manuscript.

Line 182 – Fig 2 should be referenced earlier in this paragraph. Natural herbs are not mentioned in the figure.

In accordance with your suggestion, we moved Figure 2 immediately after the description regarding NF-kB inhibitors on page6, line 217, in the revised manuscript.

Line 229 – there should not be an “s” on the word osteoid

I apologize my careless mistakes. I deleted “s” on page 8, line 325, in the revised manuscript.

The sentences beginning on lines 233 and 235 are quite redundant

In accordance with your suggestion, we added a sentence “The expression of numerous bone-related extracellular matrix proteins, and the activity of alkaline phosphatase (ALP), is key features of osteoblasts” on page 8, lines 326-327, in the revised manuscript.

There is now data on the alternative NF-kB pathway in osteoblasts that is not at all addressed in this review. Papers that should be considered include Yao JBMR 29:866; Zhang JCI 124:297; Davis JBMR 34:2087

In accordance with your suggestion, we added 2 paragraphs regarding the role of the alternative NF-kB pathway in osteoblasts from page 10, line 403, to page 11, line 429, and 4 references #96-99, in the revised manuscript.

In Figure 3, it is not clear what the words “time-saving” mean/refer to. Greek symbols seem to be missing from the references. 

In accordance with your suggestion, we replaced new Figure 3 and Figure legend, on page 10, lines 394-401, in the revised manuscript.

We forgot to mention about the importance of p65 for human skeletal homeostasis reported by Frederiksen et al., published in J Bone Miner Res 2016 31 :163-172, in the original manuscript. Since this is an important, I added a paragraph regarding the article on page 10, lines 385-393 and a reference #95 in the revised manuscript.

I hope that the revised manuscript would satisfactorily answer the comments raised by you.

Changes made with red have been written in the revised manuscript.

Reviewer 2 Report

Cells-653607

The role of NF-κB in physiological bone development and inflammatory bone diseases: Is inhibition of NF-κB “killing two birds with one stone”?

This review article summarizes the role of transcription factor NF-κB in physiological bone formation and resorption as well as pathophysiological bone destruction occurring in the course of inflammatory bone diseases. The background of NF-κB including its subunits, activators, signaling pathways, and functions is illustrated, role and characteristics of involved cell types are described, and carefully selected examples of inflammatory bone diseases are provided.

Text and figures appear straightforward and clear. The review article covers the selected topic, reflects relevant parts of the latest literature, and provides an informative overview for the reader. Therefore, only a few a few aspects should be addressed.

1. On page 2, lines 54-55, the authors state that “…, IκB is phosphorylated and ubiquitinated by the IκB kinase complex …”. This should be clarified since the ubiquitination is not directly mediated by IKK. 2. Please provide some examples for the IKKβ inhibitors (pg. 5, ln. 177) and natural herbs (pg. 5, ln. 179) mentioned. 3. A few sentences concerning the influence of nutrition on NF-κB in the context of inflammatory bone diseases should be included. 4. Please specify the specific inhibitor(s) of NF-κB (pg. 6, ln. 245 and 247; pg. 7, ln. 258 and 266) and the mode of NF-κB inhibition (pg. 6, ln. 246) mentioned. 5. Please define the abbreviation OVX (pg. 6, ln. 248). 6. To me, the paragraph describing the opposing effects of BMP and NF-κB (pg. 7, esp. ln. 256-267) is difficult to interpret. For instance, the authors describe that an “… inhibitor of NF-κB abolished … osteoblast differentiation induced by TNFα” (ln. 258-260). On the other hand, NF-κB inhibition via zinc or PDTC promotes BMP2-induced osteoblast differentiation or blocks TNF-induced suppression of osteoblast differentiation, respectively (ln. 260-262). The authors conclude that “… inhibition of NF-κB reverses osteoblast differentiation …” (ln. 263) whereas “… selective inhibitors of NF-κB have the effect of promoting bone formation … ” (ln. 266-267). In total, with respect to the effect of NF-κB (and TNF) on osteoblast differentiation and bone formation, the information provided appears inconsistent and should be clarified. 7. In the text, the possible side effects mentioned (e.g., pg. 7 ln. 267 and 282) should be addressed. 8. In the figures (at least in the version available), a variety of little squares are included that don’t seem to fulfil any function (perhaps remnants from the creation process). Please correct. 9. In the figure legends, already defined abbreviations (such as TNF, MMP) should be used instead of the full Terms.

Author Response

To Reviewer #2,

We greatly appreciate your favorable comments. We are very pleased to know that you have found our review important and interesting.

First of all, in accordance with reviewer 3’s suggestion, we added new description regarding the advantage of NF-κB inhibitors compared with either TNFablockers or IL-6 neutralized antibody for the treatment with RA, and the involvement of NF-κB on the development of ankylosing spondylitis in the revised manuscript. Since we are dentists and working on the basic research, we are not experts in these matters. Thus, from the standpoint of an orthopedic surgeon, Dr. Nakamura reviewed our description. The added part is written in blue. Since, unless Dr. Nakamura’s proper advice, we could not finish the revised manuscript, we added his name as a new co-author in the revised manuscript.

On page 2, lines 54-55, the authors state that “…, IκB is phosphorylated and ubiquitinated by the IκB kinase complex …”. This should be clarified since the ubiquitination is not directly mediated by IKK.

In accordance with your suggestion, we rewrote “ When cells are stimulated by inflammatory cytokines such as TNF-aand IL-1, IkB is phosphorylated by the IkB kinase complex (IKKa, IKKb, NEMO: NF-kB essential modulator), ubiquitinated and then degraded by the ubiquitin-proteasome system” on page 2, lines 56-59, in the revised manuscript.

Please provide some examples for the IKKβ inhibitors (pg. 5, ln. 177) and natural herbs (pg. 5, ln. 179) mentioned.

In accordance with your suggestion, we added name of IKKbinhibitors from page 5, line 206 to page 6, line 215, in the revised manuscript.

Since all reviewers pointed “natural herbs” was not suitable, we changed “plant extracts” and added some examples on page 7, lines 254-257, in the revised manuscript.

A few sentences concerning the influence of nutrition on NF-κB in the context of inflammatory bone diseases should be included.

We may not fully understand what you would like to describe, but I added the effects of ω-3 polyunsaturatedfatty acids and olive oil on RA treatment on page 7, lines257-268, in the revised manuscript.

Please specify the specific inhibitor(s) of NF-κB (pg. 6, ln. 245 and 247; pg. 7, ln. 258 and 266) and the mode of NF-κB inhibition (pg. 6, ln. 246) mentioned.

In accordance with your suggestion, we added name and target molecules of NF-κB inhibitors on page 9, line 340-343, in the revised manuscript.

Please define the abbreviation OVX (pg. 6, ln. 248).

I apologize my careless mistakes and added a word “ovariectomized” on page 9, line 343, in the revised manuscript.

To me, the paragraph describing the opposing effects of BMP and NF-κB (pg. 7, esp. ln. 256-267) is difficult to interpret. For instance, the authors describe that an “… inhibitor of NF-κB abolished … osteoblast differentiation induced by TNFα” (ln. 258-260). On the other hand, NF-κB inhibition via zinc or PDTC promotes BMP2-induced osteoblast differentiation or blocks TNF-induced suppression of osteoblast differentiation, respectively (ln. 260-262). The authors conclude that “… inhibition of NF-κB reverses osteoblast differentiation …” (ln. 263) whereas “… selective inhibitors of NF-κB have the effect of promoting bone formation … ” (ln. 266-267). In total, with respect to the effect of NF-κB (and TNF) on osteoblast differentiation and bone formation, the information provided appears inconsistent and should be clarified.

In accordance with your suggestion, we rewrote 4 sentences “Since BMP signaling and the classical NF-kB pathway have opposing biological activities, crosstalk between the two is possible. A cell-permeable inhibitor of the classical NF-kB pathway restored the inhibitory effects of TNF-aon BMP2-induced Runx2 expression and osteoblast differentiation [90]. Zinc inhibits the classical NF-kB pathway by TNF-aand promotes BMP2-induced osteoblast differentiation [91]. Pyrrolidine dithiocarbamate (PDTC), which inhibits the classical NF-kB pathway, partially blocked TNF-a-induced suppression of osteoblast differentiation induced by BMP [92]. “ on page 9,lines 355-361, in the revised manuscript.

In the text, the possible side effects mentioned (e.g., pg. 7 ln. 267 and 282) should be addressed.

In accordance with your suggestion, we modified a sentence “These results suggest that selective inhibitors of the classical NF-kB pathway have the effect of promoting bone formation by BMP, but side effects from administration must be considered, since inhibition of the classical NF-kB pathway activity might induce cell death [26-31].” on page 9, lines 365-368, in the revised manuscript.

In the figures (at least in the version available), a variety of little squares are included that don’t seem to fulfil any function (perhaps remnants from the creation process). Please correct.

I could not see any squares on all of the figures in my computer and printed both word and pdf versions. I am not sure what are these, but I converted ppt file to tif file and then attached new all of the figures in the revised manuscript.

In the figure legends, already defined abbreviations (such as TNF, MMP) should be used instead of the full Terms.

In accordance with your suggestion, we used abbreviations in the “Figure legends” section, in the revised manuscript.

We forgot to mention about the importance of p65 for human skeletal homeostasis reported by Frederiksen et al., published in J Bone Miner Res 2016 31 :163-172, in the original manuscript. Since this is an important, I added a paragraph regarding the article on page 10, lines 385-393 and a reference #95 in the revised manuscript.

I hope that the revised manuscript would satisfactorily answer the comments raised by you.

Changes made with red have been written in the revised manuscript.

Reviewer 3 Report

The review article by Jimi et al., is a timely and concise review of the role of NF-kB in bone disease. The manuscript primarily focuses NF-κB signalling and how this is involved in bone resorption and inflammatory bone diseases. Overall, the article is well structured and clearly written, however, I do have a few suggestions on how the article for improvement.

Can the authors comment whether the genes induced by p50/p65 or p52/RelB are the same or different? Here, I would suggest to modify Figure 1 to indicate two different sets of target genes, one induced by p50/p65 and the other by p52/RelB and give examples.

Lines 151-152. The following sentence is not completely clear; “stimulated with RANKL do not cause p100 to p52 processing”. Please rephrase.

IL-6 inhibitors (e.g. tocilizumab/ ACTEMRA) are also commonly used in the treatment of rheumatoid arthritis, please discuss how these may also influence NF-κB signalling.

The section discussing “natural herbs” is very short and not as well formed as the rest of the article. Furthermore, no references or explanations are given for the statement “Natural herbal ingredients are considered less likely to have side effects than NF-kB inhibitors”. Finally, Figure 2 corresponds to NF-κB inhibitors in general and not natural herbal ingredients. Please improve these sections.

Please also comment in the article why Nf-Kb inhibitors should be better than TNFa blockers.

I would also like a small section on Ankylosing spondylitis to be included. Here the one of the best treatments are TNF-a blockers, which implies a role for NfKb in this disease. Perhaps this is another disease that could benefit from Nf-Kb inhibitors?

Figure 3 is much lower in quality compared to the rest of the article. Please improve.

Minor

Please spell out BMP upon first use.

The authors have tended to use “On the other hand” quite frequently, I would suggest to change one or two of these occurrences with “while” or “conversely” etc.

Author Response

To Reviewer #3,

We would like to thank you for your suggestions, which greatly enhanced our revised manuscript.

First of all, in accordance with your suggestion, we added new description regarding the advantage of NF-κB inhibitors compared with either TNFablockers or IL-6 neutralized antibody for the treatment with RA, and the involvement of NF-κB on the development of ankylosing spondylitis in the revised manuscript. Since we are dentists and working on the basic research, we are not experts in these matters. Thus, from the standpoint of an orthopedic surgeon, Dr. Nakamura reviewed our description. The added part is written in blue. Since, unless Dr. Nakamura’s proper advice, we could not finish the revised manuscript, we added his name as a new co-author in the revised manuscript.

Can the authors comment whether the genes induced by p50/p65 or p52/RelB are the same or different? Here, I would suggest to modify Figure 1 to indicate two different sets of target genes, one induced by p50/p65 and the other by p52/RelB and give examples.

So far, unfortunately, we could not any comments the difference of target genes induced by p50/p65 or p52/RelB. I added 2 sentences “ Since these two pathways play different roles, the p50/p65, p50/c-Rel and the p52/RelB heterodimers are expected to bind to their specific DNA sequences. However, the sequence to which p52/RelB heterodimer specifically binds, and it’s target genes have not been identified [11,12]” and 2 references #11, 12, in the revised manuscript.

Lines 151-152. The following sentence is not completely clear; “stimulated with RANKL do not cause p100 to p52 processing”. Please rephrase.

In accordance with your suggestion, we modified a sentence “Osteoclast progenitor cells derived from NIK-/-mice did not induce processing of p100 to p52 by RANKL stimulation due to IkB-like function of C-terminus of p100 [39,40].” On page 4, lines 161-162, in the revised manuscript.

IL-6 inhibitors (e.g. tocilizumab/ ACTEMRA) are also commonly used in the treatment of rheumatoid arthritis, please discuss how these may also influence NF-κB signalling.

In accordance with your suggestion, we added 6 sentences “Recently biological products, such as anti-TNF- neutralized antibody (etanercept, infliximab, and adalimumab, e.t.c.,) and anti-IL-6 neutralized antibody (tocilizumab), are drugs created by biotechnology technology, and have been used for rheumatoid arthritis. Compared to conventional anti-rheumatic drugs, the cost is high, but it is known to be particularly effective in suppressing joint destruction. The treatment guidelines that prevent the destruction of joints by introducing biologics as soon as possible when treatments centered on rheumatox are not enough to control the disease are widely accepted internationally [47]. Anti-TNF- neutralized antibodies directly inhibit the binding of TNF- to its receptor and suppress excessive inflammation that induces RANKL expression in synovial. IL-6 is required for the differentiation of Th17 cells that promote osteoclast differentiation, and these neutralizing antibodies are thought to not only sink local inflammation but also suppress RANKL induction and osteoclast differentiation. However, these biologics causes serious side effects such as triggering an autoimmune anti-antibody response or the weakening of the body's immune defenses. Therefore, alternative small-molecule based therapies for inhibition of these cytokines’ effects is a hot topic both in academia and industry [47,48].” on page 5, lines 189-203, and 2 references #47, 48, in the revised manuscript.

The section discussing “natural herbs” is very short and not as well formed as the rest of the article. Furthermore, no references or explanations are given for the statement “Natural herbal ingredients are considered less likely to have side effects than NF-kB inhibitors”. Finally, Figure 2 corresponds to NF-κB inhibitors in general and not natural herbal ingredients. Please improve these sections.

Since all reviewers pointed “natural herbs” was not suitable, we changed “plant extracts” and added some examples on page 7, lines 254-257, in the revised manuscript. We also moved Figure 2 immediately after the description regarding NF-kB inhibitors on page6, line 217, in the revised manuscript.

Please also comment in the article why NF-kB inhibitors should be better than TNF-ablockers.

In accordance with your suggestion, as describe above Q3, we combined bot IL-6 antibody and TNF-ablockers. We added 6 sentences on page 5, lines 189-203, and 2 references #47, 48, in the revised manuscript.

In accordance with Dr. Nakamura’s suggestion, we added a paragraph thet iguratimod, originally generated as an NF-kB inhibitor, is now being widely used for RA treatment, from page 9, line 242 to page 7, line 253, and 3 references #55-57, in the revised manuscript.

I would also like a small section on Ankylosing spondylitis to be included. Here the one of the best treatments are TNF-a blockers, which implies a role for NF-kB in this disease. Perhaps this is another disease that could benefit from NF-kB inhibitors?

In accordance with your suggestion, we added a paragraph from page 7, line 279 to page 8, line 301, and 4 references #73-76, in the revised manuscript.

Figure 3 is much lower in quality compared to the rest of the article. Please improve.

In accordance with your suggestion, we replaced new Figure 3 and Figure legend, on page 10, lines 394-401, in the revised manuscript.

Minor

Please spell out BMP upon first use.

I apologize my careless mistakes and added a word “Bone morphogenetic proteins” on page 9, line 353, in the revised manuscript.

The authors have tended to use “On the other hand” quite frequently, I would suggest to change one or two of these occurrences with “while” or “conversely” etc.

In accordance with your suggestion, we changed “On the other hand”to “While”, “By contrast” or “Whereas” in the revised manuscript.

We forgot to mention about the importance of p65 for human skeletal homeostasis reported by Frederiksen et al., published in J Bone Miner Res 2016 31 :163-172, in the original manuscript. Since this is an important, I added a paragraph regarding the article on page 10, lines 385-393 and a reference #95 in the revised manuscript.

I hope that the revised manuscript would satisfactorily answer the comments raised by you.

Changes made with red have been written in the revised manuscript.